American Society for Microbiology | Microbiology Spectrum

# Berberine Depresses Inflammation and Adjusts Smooth Muscle to Ameliorate Ulcerative Colitis of Cats by Regulating Gut Microbiota

Xueying Li,[a] Shuang Xu,[a] Yanhe Zhang,[a] Kan Li,[a] Xue-Jiao Gao,[a] Meng-yao Guo[a]

[a]Department of Clinical Veterinary Medicine, College of Veterinary Medicine, Northeast Agricultural University, Harbin, People's Republic of China

**ABSTRACT** Intestinal microbiota dysbiosis is a well established characteristic of ulcerative colitis (UC). Regulating the gut microbiota is an effective UC treatment strategy. Berberine (BBR), an alkaloid extracted from several Chinese herbs, is a common traditional Chinese medicine. To establish the efficacy and mechanism of action of BBR, we constructed a UC model using healthy adult shorthair cats to conduct a systematic study of colonic tissue pathology, inflammatory factor expression, and gut microbiota structure. We investigated the therapeutic capacity of BBR for regulating the gut microbiota and thus work against UC in cats using 16S rRNA genes amplicon sequencing technology. Our results revealed that dextran sulfate sodium (DSS)-induced cat models of UC showed weight loss, diarrhea accompanied by mucous and blood, histological abnormalities, and shortening of the colon, all of which were significantly alleviated by supplementation with BBR. A 16S rRNA gene-based microbiota analysis demonstrated that BBR could significantly benefit gut microbiota. Western blot, quantitative PCR, and enzyme-linked immunosorbent assays (ELISAs) showed that in DSS-induced cat models, the expression of the inflammatory factors was increased, activating the JAK2/STAT3 signaling pathway, and treatment with BBR reversed this effect. The myosin light chain (MLC) phosphorylation in the smooth muscle of the intestines is associated with motility of inflammation-related diarrhea in cats. This study used gut flora analyses to demonstrate the anti-UC effects of BBR and its potential therapeutic mechanisms and offers novel insights into the prevention of inflammatory diseases using natural products.

**IMPORTANCE** Ulcerative colitis (UC) is common in clinics. Intestinal microbiota disorder is correlated with ulcerative colitis. Although there are many studies on ulcerative colitis in rats, there are few studies on colitis in cats. Therefore, this study explored the possibility of the use of BBR as a safe and efficient treatment for colitis in cats. The results demonstrated the therapeutic effects of BBR on UC based on the state of the intestinal flora. The study found BBR supplementation to be effective against dextran sulfate sodium (DSS)-induced colitis, smooth muscle damage, and gut microbiota dysbiosis.

**KEYWORDS** gut microbiome, ulcerative colitis, intestinal smooth muscle, host-bacterial interactions

Ulcerative colitis (UC), the most common type of inflammatory bowel disease (IBD), is a global public health concern (1). It develops via the convergence of environmental, microbial immunological, and genetic factors (2). As a primary form of IBD, UC mainly involves the rectum, extends to the entire colon, and affects the colonic mucosa and submucosa (3). Maintaining the structure and producing peristaltic and segmental movements of the smooth muscle is essential for the intestine (4). Intestinal microbiota disorder is an important risk factor for UC (5, 6). In recent years, increasing evidence has indicated that intestinal flora plays a vital role in the progression of colitis (7–9).

Address correspondence to Xue-Jiao Gao, xuejiaogao@126.com, or Meng-yao Guo, gmy@neau.edu.cn.

The authors declare no conflict of interest.

Thus, the reshaping of the intestinal microflora is a potential target in UC treatment intervention strategies.

Antibiotics, as the primary clinical treatment for the intestinal inflammation, may lead to disorders involving the intestinal flora of the organism and may generate drug-resistant genes, making the prevention and treatment of the disease more difficult (10). Berberine (BBR), as one of the most common traditional Chinese medicines, is an alkaloid extracted from several Chinese herbs. It has been widely used as an antidiarrheal medication and an effective remedy for metabolic disorders (11) and also has substantial antioxidant, antiinflammatory, antihyperglycemic, and hypolipidemic effects (12–15). Several studies have shown that BBR has a significant effect on the bioavailability of nutrients in the intestine, which suggests that it may impact on the gut microbiome (16–18). Studies have shown that BBR attenuates dextran sulfate sodium (DSS)-induced inflammation in rats by downregulating the JAK2/STAT3 pathway (19–21). NLRP3, an essential component of innate immunity, is activated and can regulate the caspase-1 activity, contributing to the activation of cytokine precursor pro-interleukin-1$\beta$ (pro-IL-1$\beta$) and pro-IL-18 in DSS-induced models (22–24). The mechanisms responsible for the link between the beneficial effects of BBR on colitis and gut microbiota are not fully understood. Although many previous studies have shown that BBR caused alterations in the intestinal flora of mice with inflammation (25–28), intestinal infections in cats are more common and receive an increasing degree of attention in clinical practice, which increases the value of studying intestinal diseases in cats. Dextran sulfate sodium (DSS) has been used widely to generate an experimental model of UC disease. The clinical phenotype of the animal model shares a high similarity with that of patients with UC (5, 29–31).

Our study investigated the effects of BBR on the interactions between the colon and the gut microbiota in DSS-induced cats. Comprehensive metagenomics analyses were employed to analyze the potential for regulating the gut microbiome with BBR. We used 16S rRNA gene sequencing to detect the alterations of microbiota and recognize differential metabolites in order to illuminate the mechanism of BBR in the treatment of DSS-induced cat models. Our results showed that BBR supplementation improves the intestinal barrier, restores gut microbiota, modifies the metabolic profile, and suppresses the JAK2/STAT3 signaling pathway. In addition, BBR ameliorates intestinal inflammation by restoring myosin light chain (MLC) phosphorylation in the intestinal smooth muscles to normal levels. Accordingly, our data suggest that BBR has significant potential to alleviate UC.

## RESULTS

**BBR attenuated DSS-induced colitis.** Our results showed that the induction of DSS caused typical UC symptoms in cats, such as diarrhea and hematochezia (Fig. 1B), and a marked shift in disease activity index (DAI) score (Fig. 1D) and body weight (Fig. S1). Additionally, the colon length of the cat was significantly shortened in the DSS-induced group (Fig. 1C). Hematoxylin and eosin (HE) staining of the colon showed that DSS treatment caused severe enteric mucosal injury (Fig. 1E). However, all characteristic features were prevented by oral BBR supplementation. These results indicate that BBR can alleviate the overall symptoms of DSS-induced colitis in cats and alleviate colonic injury.

**Regulation by BBR in intestinal flora in DSS-induced UC cats.** Species accumulation curves were shown in order to judge whether the sample numbers were sufficient (Fig. S2). Next, we analyzed the differences in intestinal flora at the family and genus levels (Fig. 2A and B). At the family level, *Prevotellaceae, Lachnospiraceae, Selenomonadaceae, Veillonellaceae,* and *Lactobacillacrae* were significant in the control group, and in the DSS-induced group, there was an increase in the richness of *Bacteroidaceae* and *Fusobacteriaceae*. After treatment with BBR, there was an increase in beneficial bacteria, such as *Lactobacillacrae* and *Prevotellaceae* and a reduction in the richness of *Bacteroidaceae*. At the genus level, the DSS-induced group showed an increase in the richness of *Bacteroides* and *Fusobacterium*, and after treatment with BBR, there was an increase in beneficial bacteria, such as *Lactobacillus* and

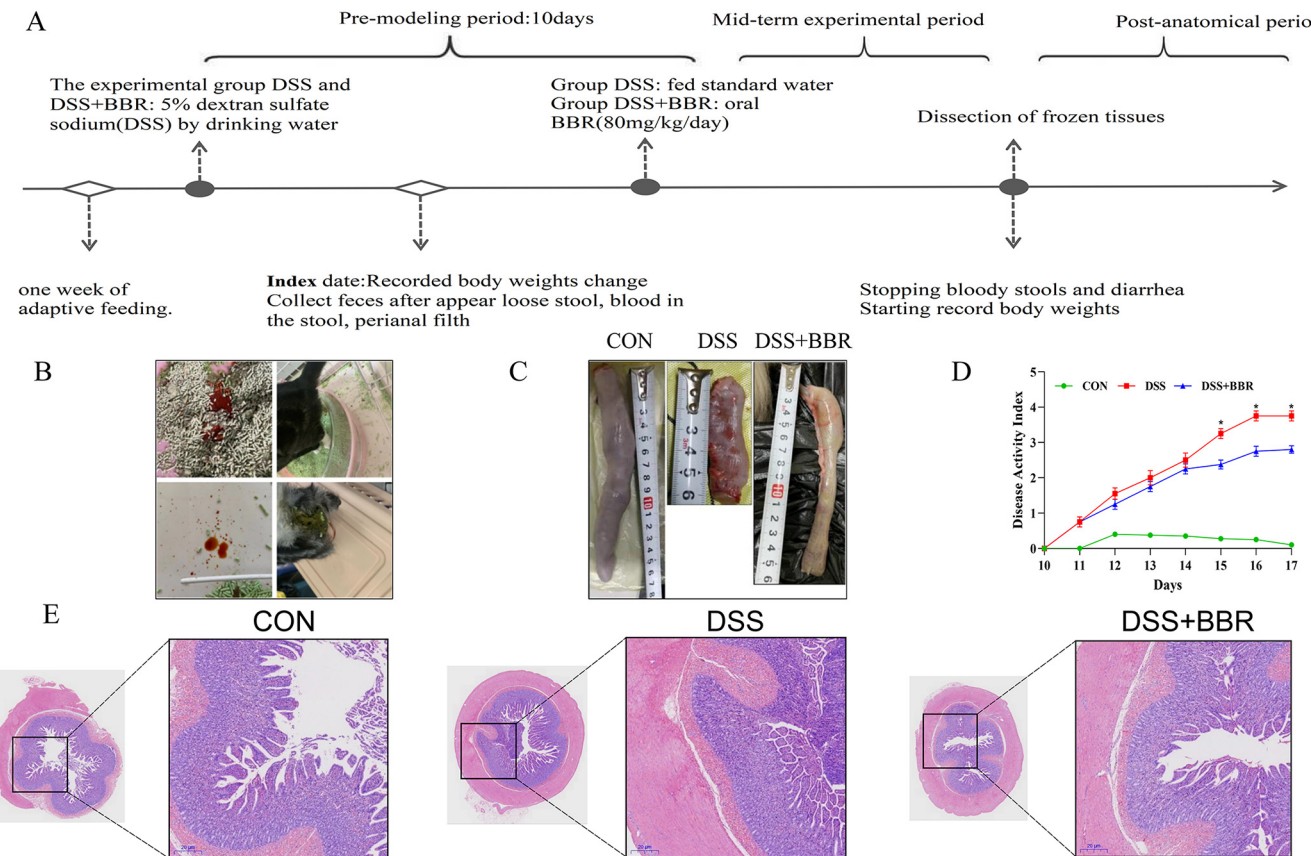

**FIG 1** Berberine (BBR) attenuates the symptoms of dextran sulfate sodium (DSS)-induced cat colitis. (A) Experimental design to test the effects of BBR on DSS-induced cat. (B) Diarrhea and blood conditions of DSS-induced colitis (DSS) group and berberine treatment (DSS+BBR) group. (C) Effect of DSS and BBR on colonic length. (D) Disease activity index of cat during colitis. (E) Representative images of hematoxylin and eosin (HE)-stained colon tissue samples. The data are expressed as means ± standard error of the mean (SEM) ($n = 4$). *, $P < 0.05$; **, $P < 0.01$.

*Prevotella* and a reduction in the richness of *Bacteroides*. The results showed that the intestinal flora of DSS-induced colitis is changed by BBR treatment.

**Effects of BBR on intestinal flora diversity of cats.** $\beta$-Diversity analysis is mainly used to analyze differences between the groups. Using a principal-component analysis (PCA) and a principal coordinates analysis (PCoA), differences in species composition abundance were compared by analyzing the projected distances on the axes among the samples. PCA (Fig. 3A) and PCoA analyses (Fig. 3B) were used to evaluate the similarities and differences between the three groups. PCA indicated that intestinal flora diversity was higher in the DSS-induced group than in the control group and was reduced after treatment with BBR. PCoA indicated that the gut microbiota composition of the BBR group was different from the DSS group in terms of axis PCo-1. These results suggest that the administration of BBR maintains the intestinal flora diversity.

**BBR effected species differences and metabolic pathway function.** In this study, we performed 16S rRNA gene high-throughput sequencing to reveal the impact of BBR on the gut microbiota. The Venn diagram is used as a community analysis to investigate which species are common and unique among different groups. It can be seen that the difference in operational taxonomic unit (OTU) between the DSS+BBR group and the DSS group at the phylum level is not apparent, mainly with *Firmicutes*, *Bacteroidetes*, and *Proteobacteria*; at the genus level, *Prevotella* shows the most evident change, followed by *Bacteroides* and *Clostridium*. There is a difference in relative abundance between the DSS+BBR group and the DSS group in the *Firmicutes*, *Fusobacteria*, and *Proteobacteria*. At the genus level, significant differences can be seen in *Prevotella*, *Bacteroides*, *Blautia*, *Cellulosilyticum*, and *Fusobacterium* in both the DSS+BBR and DSS groups (Fig. 4A and B).

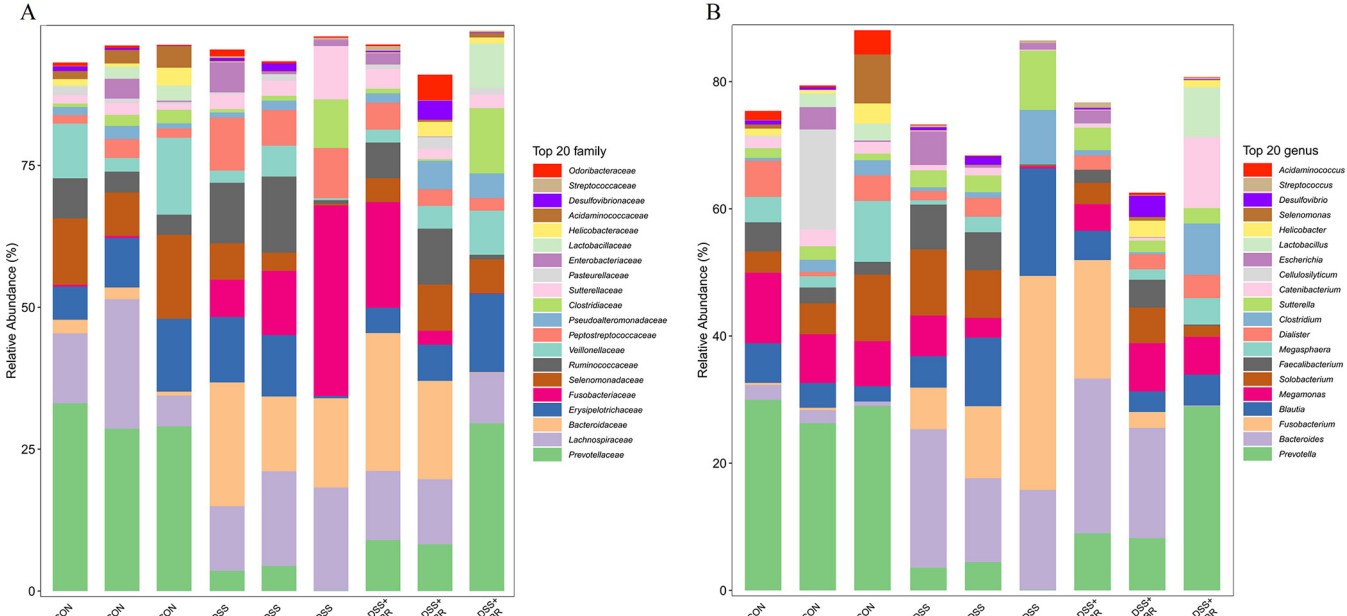

**FIG 2** Comparative structural and species analysis of gut microbiota in BBR cats. (A, B) Taxonomic analysis of microbiota in fecal samples at the top 20 family levels (A) and genus levels (B) between control group (CON), DSS-induced colitis (DSS) group, and berberine treatment (DSS+BBR) group.

Heat maps were used to analyze species composition so as to display the trend of species abundance distribution among samples and further compare species composition differences (Fig. 5A). The *Collonsella aerofaciens*, *Bifidobacterium adoiescentis*, *Megasphaera eisdenii*, *Bacteroides plebeius*, and *Lactobacillus ruminis* were significant in the control group. The DSS-induced group showed an increase in *Bacteroides fragilis* and a reduction in *Lactobacillus ruminis*. After the treatment with BBR, there was an increase in the beneficial bacteria, such as *Lactobacillus ruminis*. A random forest analysis was used to classify the microbial community samples (Fig. 5B). The DSS-induced group showed an increase in harmful bacteria, such as *Bacteroides fragilis*, and a reduction in the beneficial bacteria, such as *Lactobacillus ruminis*. However, the balance between harmful and beneficial bacteria was restored after treatment with BBR.

The analysis of fecal flora metabolic pathways in cats allowed for predicting the relevant functions of their flora and analyzing the effect of BBR on the role of feline intestinal flora. The abundance of secondary functional pathways for all samples is shown in Fig. 6A, which shows that the highest possible of metabolism-related tracks was observed and that human disease and genetic information processing were followed. Fig. 6B shows multiple MetaCyc secondary functional pathways for all samples, which demonstrates the highest quantity of ways related to biosynthesis and degradation/utilization/assimilation. These results indicate that metabolic pathway function and species differences are improved after treatment with BBR.

**BBR improved the symptoms of DSS-induced UC in cats through the JAK2/STAT3 signaling pathways.** Next, we assessed the effects of BBR administration on DSS-induced proinflammatory responses. The DSS-induced group had significantly increased serum IL-1$\beta$, IL-6, and tumor necrosis factor-$\alpha$ (TNF-$\alpha$) levels compared with the control group. Conversely, BBR supplementation remarkably reversed this tendency (Fig. 7A). Exploring the underlying mechanism by which BBR modulates DSS-induced colitis, we found that the DSS-induced group showed significant increase in mRNA level expression of inflammatory factors. The results showed that the JAK2/STAT3 signaling pathways were associated with inflammation, such as IL-6, TNF-$\alpha$, IL-1$\beta$, IL-18, NLRP3, Caspase-1, JAK2, and STAT3 (Fig. 7B). As seen in Fig. 7C, the protein expressions of JAK2, STAT3, and other related proteins in the pathway were increased in the DSS group

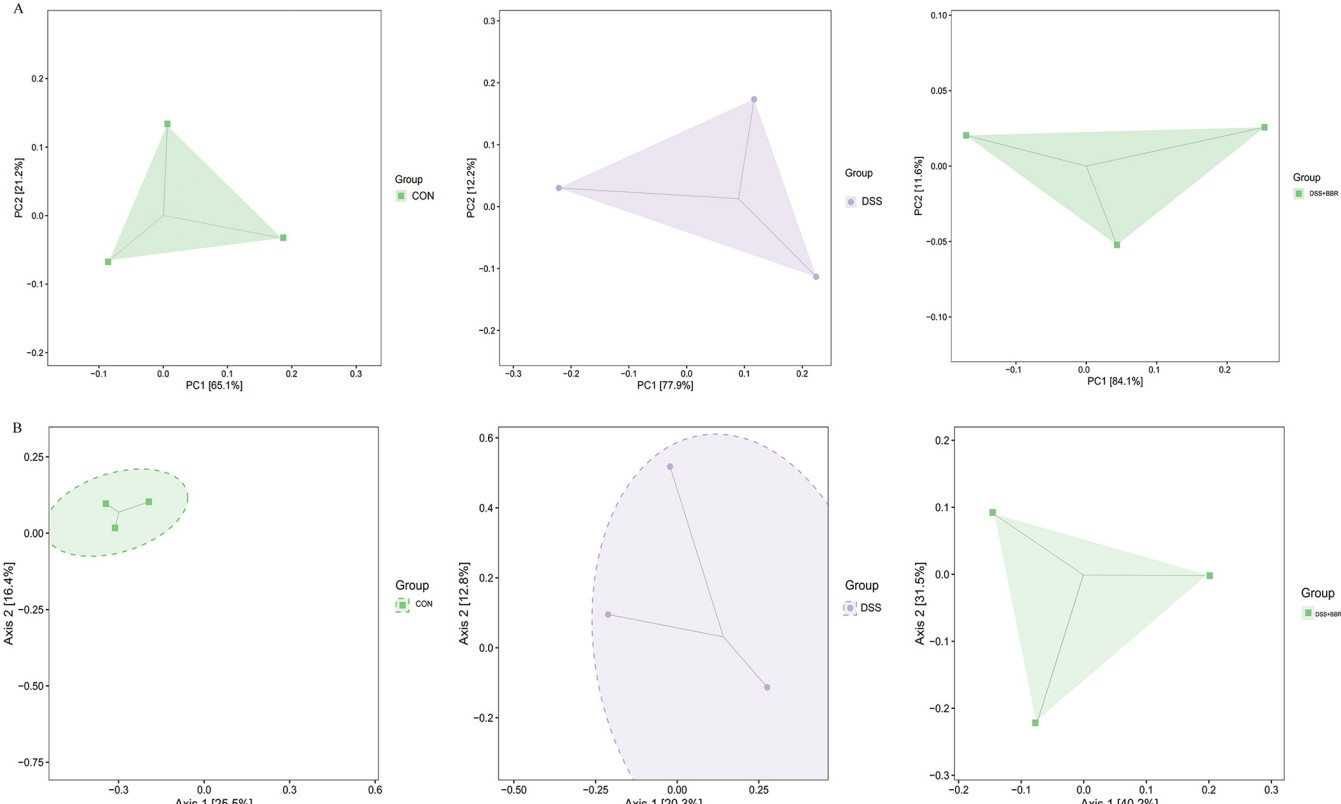

**FIG 3** Effects of BBR on intestinal flora diversity of cats. $\beta$-Diversity was assessed using. (A) PCA, (B) PCoA. Every point in the figure represents a sample, and points of different colors indicate different groups. The percentages of the coordinate axes are shown the influences that cause differences in species composition between groups.

compared to the control group. The protein expressions in the BBR group were decreased compared to those in the DSS group. Still, their expressions were higher than in the control group. The above results demonstrate that BBR can reduce inflammation through the JAK2/STAT3 signaling pathway.

**BBR leads to contraction of colonic smooth muscle.** Our study showed that the $Ca^{2+}$ concentration (Fig. 8A) was changed with the intestinal smooth muscle damage. DSS-induced models showed increases in $Ca^{2+}$ concentration, while the results were the opposite in the BBR group. Intestinal motility is positively associated with MLC phosphorylation in the smooth muscle. The relative activities of MLCK regulate smooth muscle MLC phosphorylation. The present study results indicated a significant increase in MLC phosphorylation in the colonic smooth muscle of the DSS-induced cats (Fig. 8B and C). However, these effects were significantly reversed after the BBR administration. The increased MLCK mRNA expression levels also changed considerably (Fig. 8D). It is thus concluded that BBR has a substantial impact on colonic smooth muscle contraction.

## DISCUSSION

IBD is a common chronic disease receiving global public health research attention (32). UC is an atopic form of IBD with frequent morbidity. The development of colitis is accompanied by changes in the contraction of the smooth muscle of the colon. In recent years, many studies have focused on the relationship between UC and related diseases and intestinal flora (33–35). The functions of the intestinal flora include aiding in the digestion of food, breakdown and absorption of nutrients, secretion of appropriate amounts of antimicrobial peptides to defend against colonization by harmful bacteria, and maintenance of the intestinal mucosal barrier. The impact of the intestinal flora on the immune system are

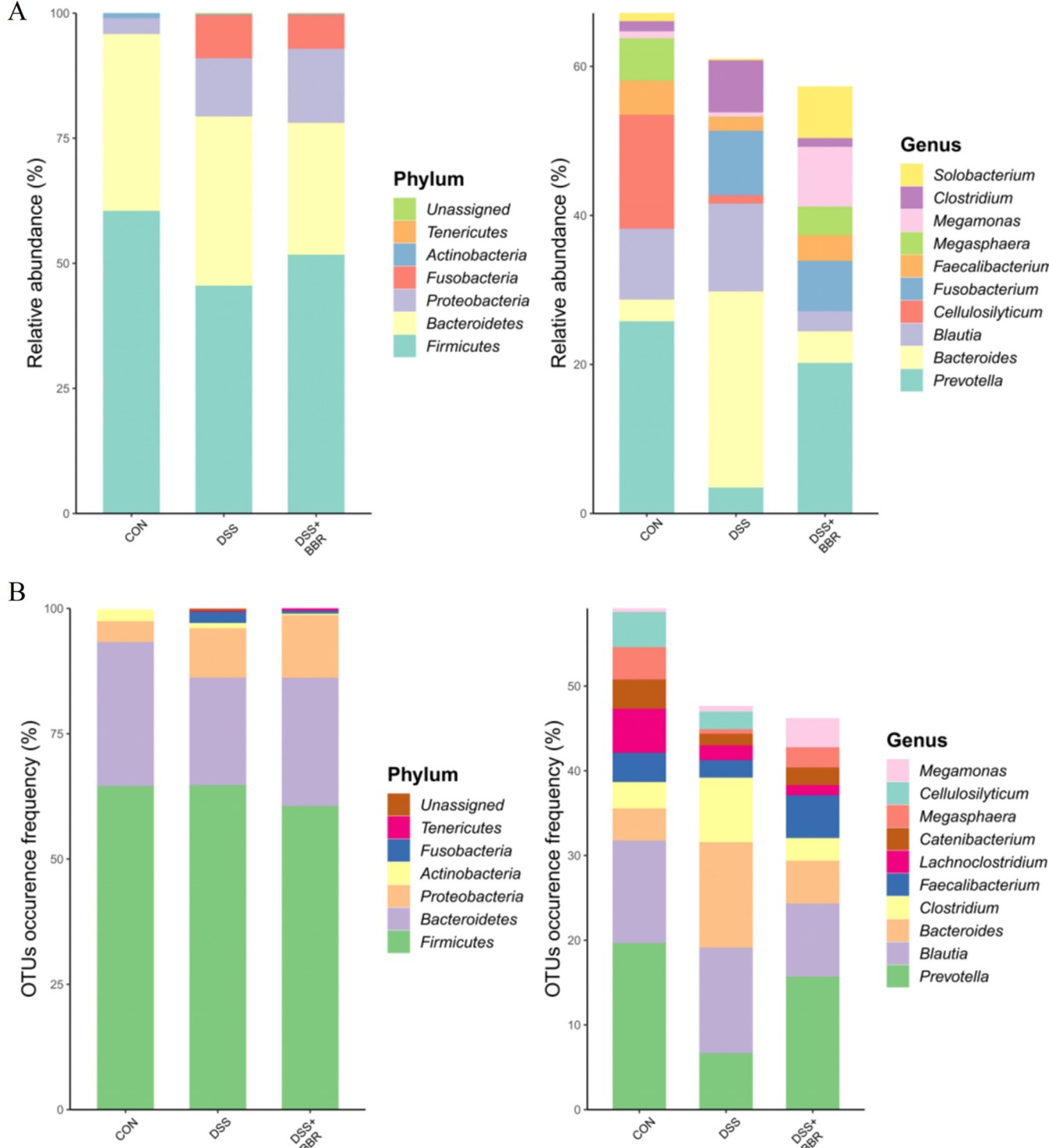

**FIG 4** Community analysis of DSS-induced colitis (DSS) group and berberine treatment (DSS+BBR) group. (A) Statistical histogram of proportion of relative abundance in phylum and genus of the Wayne chart. (B) Venn diagram of sample group amplicon sequence variants (ASV)/operational taxonomic unit (OTU) between three groups.

also important (36, 37). Our studies used DSS to induce UC in model cats because it can successfully induce high rates of stable colitis-associated characteristics. As expected, the cats with DSS-induced colitis showed various disease features comparable to those observed in patients with UC. The use of antibiotics and immune-suppressive drugs for colitis has become common, but most of these treatments have side effects (38,

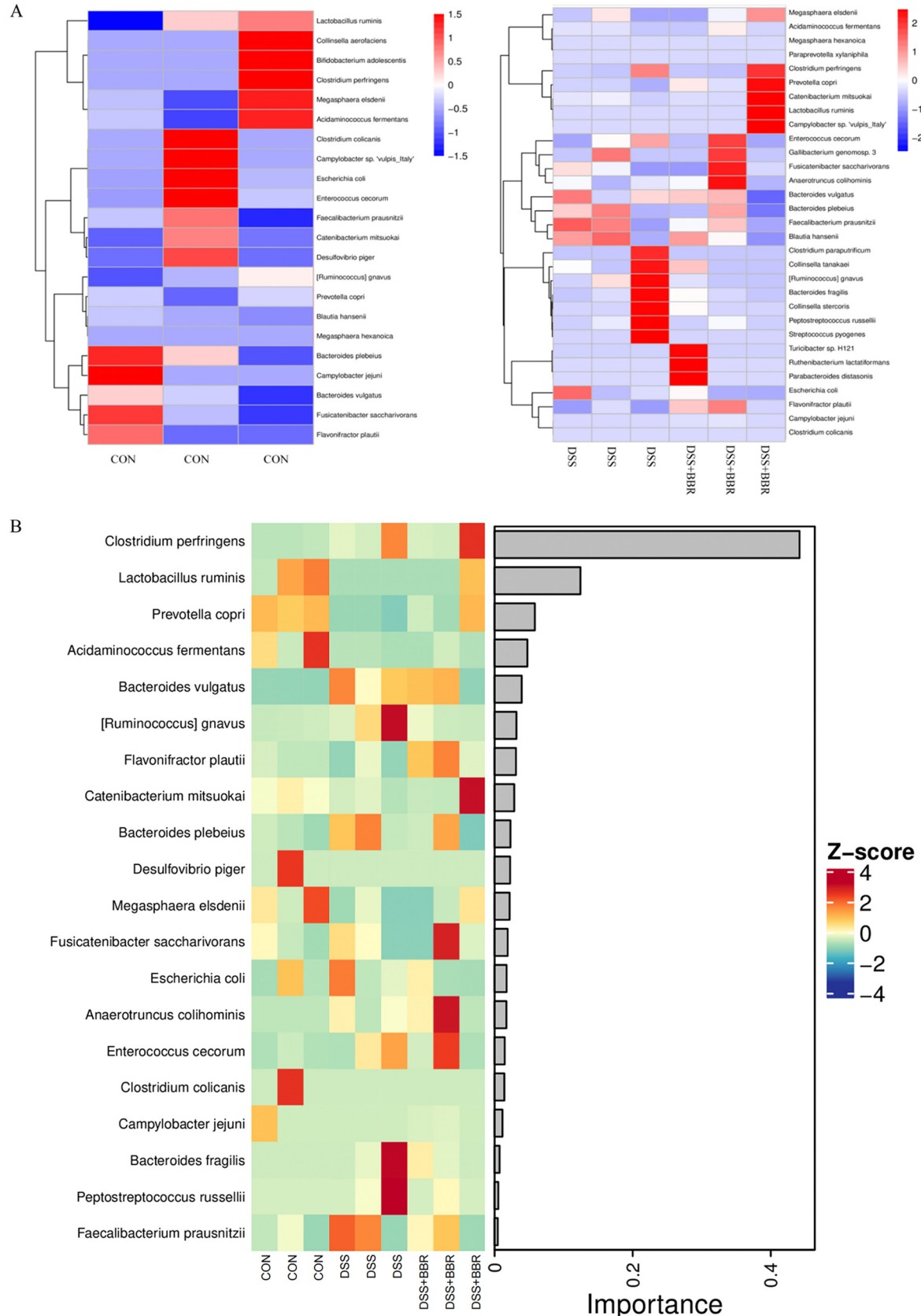

**FIG 5** BBR causes species difference analysis and marker species. (A) Genus level species composition heat map of species. (B) Random forest analysis of the dominant biomarker taxa between three groups.

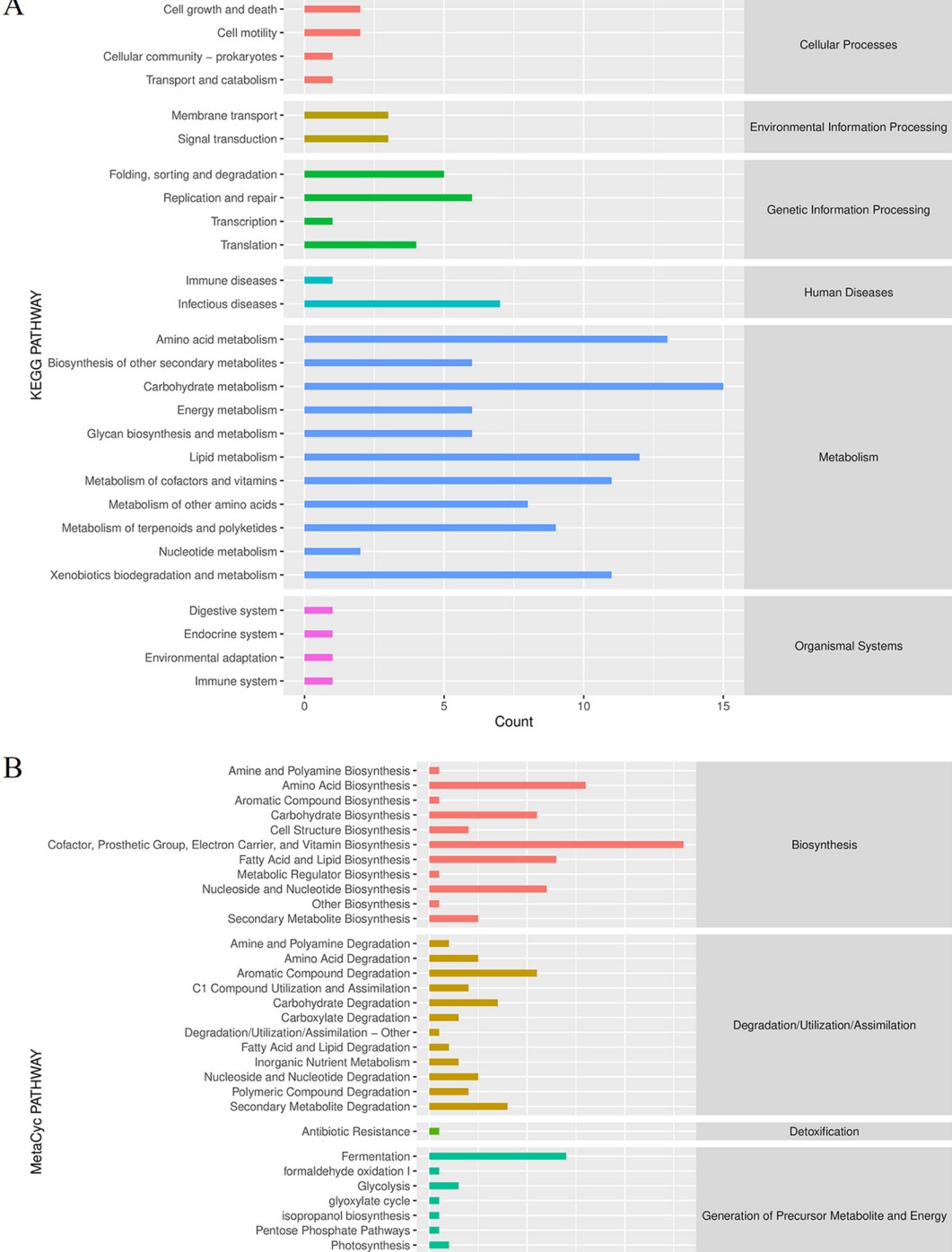

**FIG 6** BBR causes altered metabolic pathway function. (A, B) Predicted KEGG (A) and MetaCyc (B) secondary functional pathway analysis of DSS-induced colitis (DSS) group and berberine treatment (DSS+BBR) group.

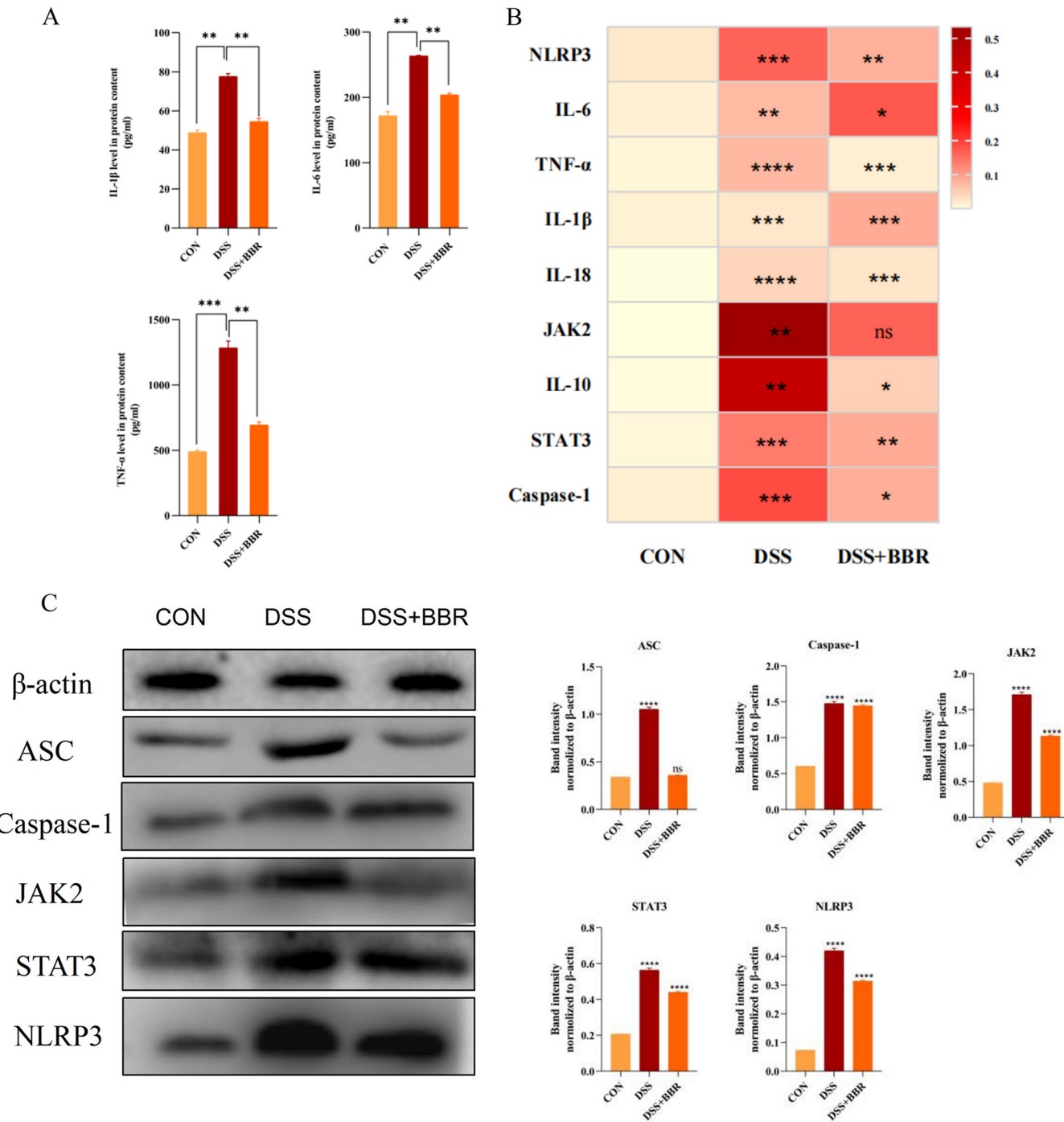

**FIG 7** Effects of BBR on inflammatory-associated factors in DSS-induced cats. (A) Interleukin-1$\beta$ (IL-1$\beta$), IL-6, and tumor necrosis factor-$\alpha$ (TNF-$\alpha$) were detected using enzyme-linked immunosorbent assay (ELISA) kits. (B) Relative mRNA expression of inflammatory factors in the colon evaluated by quantitative reverse transcription PCR (qRT-PCR). (C) JAK2, STAT3, and other related proteins in the pathway protein expression. The data are expressed as the means $\pm$ SEM ($n = 4$) and analyzed using one-way analysis of variance (ANOVA) with Tukey *post hoc* analysis. DSS-induced colitis (DSS) group/ berberine treatment (DSS+BBR) group (versue control group [CON]). *, $P < 0.05$; **, $P < 0.01$; ***, $P < 0.001$; ****, $P < 0.0001$.

39). As an alternative, traditional plant therapy has received increasing attention. BBR, as an over-the-counter medicine, is readily available and inexpensive. Many studies have investigated the mechanism of action of herbal medicine in the treatment of colitis (40–42). Our study found BBR supplementation to be effective against DSS-induced colitis, smooth muscle damage, and gut microbiota dysbiosis.

Pathogenic intestinal microbiota plays a critical role in chronic inflammation. The

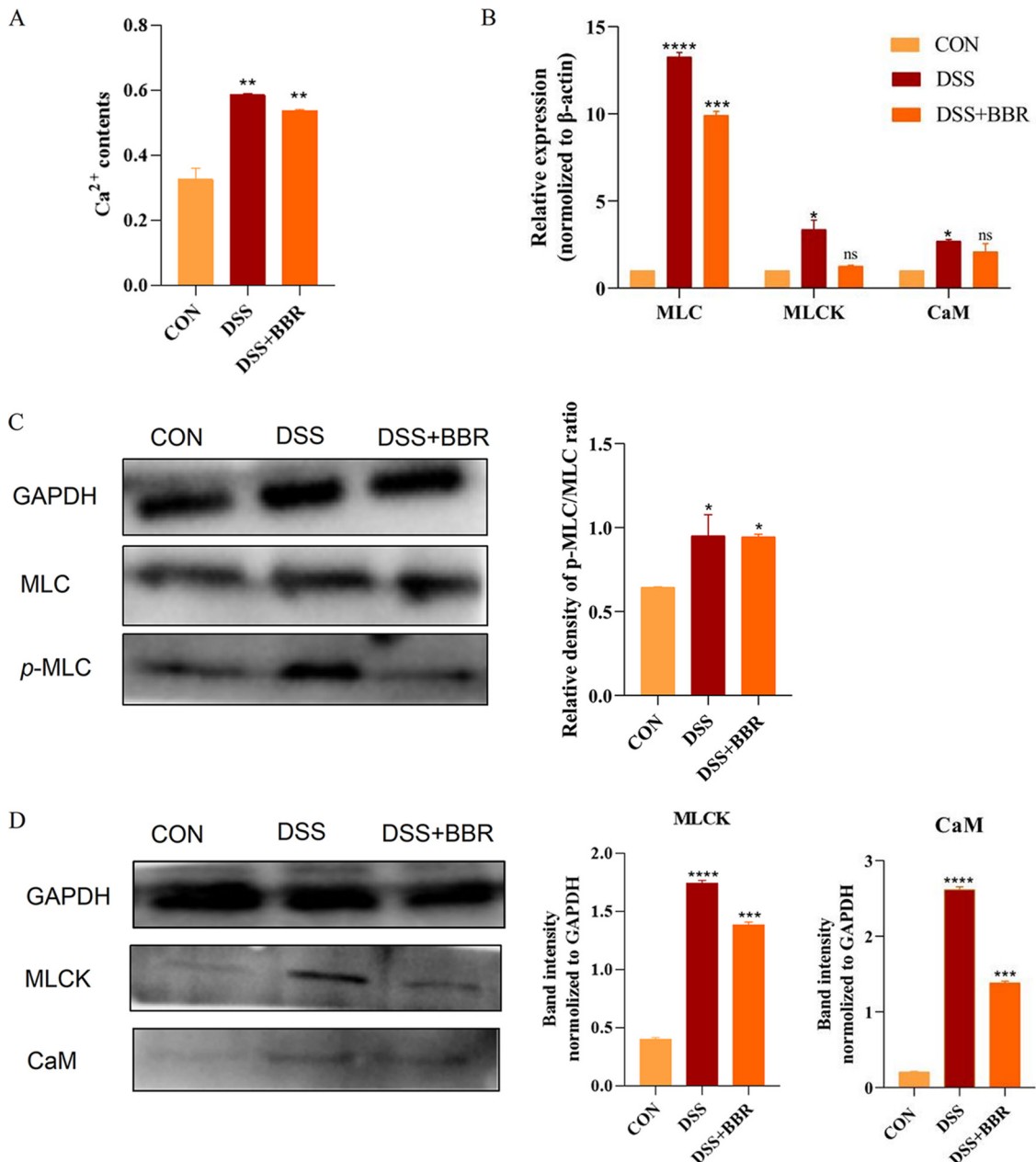

**FIG 8** Alterations of distal colon smooth muscle contraction. (A) Effect of BBR on Ca$^{2+}$ content in colon smooth muscle contraction. (B) Relative mRNA expression of smooth muscle factors in the colon evaluated by qRT-PCR. (C, D) The protein expression of myosin light chain (MLC), MLC phosphorylation (p-MLC), p-MLC/MLC, MLCK myosin light chain kinase (MLCK), and calmodulin (CaM) in BBR-treated DSS cat models. The data are expressed as the means ± SEM (n = 4) and analyzed using one-way ANOVA with Tukey *post hoc* analysis. DSS-induced colitis (DSS) group/berberine treatment (DSS+BBR) group (versue control group [CON]). *, $P < 0.05$; **, $P < 0.01$; ***, $P < 0.001$; ****, $P < 0.0001$. GAPDH, glyceraldehyde-3-phosphate dehydrogenase.

alteration of the gut environment can lead to microbiota dysbiosis. Intestinal microbial composition and diversity are decreased in mice with colitis (43–45). DSS changed the colon microbial community composition and diversity. The development of ulcerative colitis leads to a disturbance in the gut microbiota; resident bacteria are excreted in large numbers through the feces during diarrhea. For instance, with the highest relative abundance of microbiota, *Bacteroidetes* was increased and *Firmicutes* was decreased by DSS. The increase of *Bacteroidetes* and the ratio of *Bacteroidetes*/*Firmicutes* are the indicators of IBD (17, 46, 47). Our results showed that BBR can decrease *Bacteroidetes* and increase *Firmicutes*

in DSS-induced cats. BBR reshaped the microbiota composition by reducing the abundance of *Proteobacteria*. In the genus of bacteria, BBR can regulate the biodiversity and structure of the intestinal flora of cats. The richness and homogeneity of species is increased with the increase of beneficial bacteria and the decrease of harmful bacteria in the organism, such as *Prevotellaceae*. In the treatment of DSS-induced colitis with BBR, restoration of the mucosal barrier and the extent of the inflammatory response is improved by regulation of the intestinal flora.

At the biomolecular level, BBR causes a reduction of inflammatory damage in the animal organism by activating inflammatory vesicle NLRP3, which binds to ASC. Further, the expression of inflammatory factors is decreased, such as IL-6, IL-18, IL-10, TNF-$\alpha$, and IL-1$\beta$ (14, 48). IL-6 activates the downstream JAK2. The overexpression of IL-10 in UC mice may be due to an imbalance between proinflammatory and antiinflammatory cytokines due to insufficient secretion of antiinflammatory factors, and BBR causes changes in inflammatory factors through the JAK2/STAT3 signaling pathway (49). Consistent with previous investigations, our results showed that BBR can inhibit inflammatory-associated mediators' expression by dampening the JAK2/STAT3 signaling activation in cats with DSS-induced colitis. The present study results demonstrate that BBR improves the symptoms and inflammation in DSS-induced cats, which lays the foundation for further investigation of the effects of BBR on smooth muscle motility-related phosphorylation (50). The phosphorylation of MLC regulates smooth muscle contractility. Motility disorders related to abnormal regulation of the Ca$^{2+}$ are observed in gastrointestinal inflammatory disorders (51, 52). As in previous studies, our results showed that the increasing MLC phosphorylation in the DSS-induced cats was returned to normal levels by treatment with BBR, indicating that BBR benefits inflammation-related intestinal motility. BBR can restore intestinal smooth muscle function.

**Conclusion.** Our study investigated the therapeutic effects of BBR on UC based on intestinal flora and tissue distribution analysis. BBR plays a vital role in modulating intestinal flora and intestinal smoothing mechanisms by improving UC symptoms because of its antiinflammatory effects in pathological conditions. Notably, BBR tends to have unique effects that remain consistent when used in clinical practice. BBR has a long tradition as a botanical medicine for treating gastrointestinal disorders, and these data are expected to provide necessary information for the targeted application of BBR in UC treatment.

## MATERIALS AND METHODS

**Experimental protocol and animals.** Nine healthy adult shorthair cats (weight, 2.0 to 2.3 kg) were randomly allocated into three groups: control group (CON), DSS-induced colitis (DSS) group, and berberine treatment (DSS+BBR) group. All experimental studies were conducted in accordance with the humanitarian spirit and in accordance with the standards of the Institutional Animal Care and Use Committee of Northeast Agricultural University (grant number NEAUEC20220340). The cats in the DSS group and DSS+BBR group were given 5% DSS in water for 10 days. Between days 10 and 17, the DSS+BBR group was administered BBR (80 mg/kg/day), and the weight of each animal was recorded. On the day 18 of the experiment, the cats were anesthetized by intraperitoneal injection with pentobarbital sodium (35 mg/kg) before samples of tissues and blood were collected. Colon samples were harvested and stored at −80°C for further experiments. The animals were housed under standard conditions and received humane care complying with institutional guidelines. The experimental design used to test the effects of BBR on DSS-induced cats is shown in Fig. 1A.

**Disease activity index (DAI) measures.** DAI is a comprehensive score that has been used to evaluate the severity of DSS-induced colitis in animal models, including general health condition, weight loss, stool consistency, and degree of fecal bleeding (30). The scoring is as follows: score 0 indicates no weight loss, average stool properties, negative occult blood test; score 1 indicates a weight loss of 1 to 5%, soft stool, and negative occult blood test; score 2 indicates a weight loss of 5 to 10%, pale stool, and positive occult blood test; score 3 indicates a weight loss of 10 to 15%, diarrhea, and positive occult blood positive test; and 4 points indicates a weight loss of >15%, diarrhea, and bloody stools. From days 10 to 17, the general health condition, weight loss, stool consistency, and degree of fecal bleeding of the three groups was recorded.

**16S rRNA gene sequencing study and bioinformatics analysis.** Fecal samples were collected from the three groups, and the fecal microbiomes were examined using 16S rRNA gene sequencing carried out by the Shanghai Personalbio Biotechnology Corp. The total genomic DNA was extracted according to the manufacturer's protocol. The DNA concentration and purity were tested using Nanodrop and 1% agarose gel

**TABLE 1** Primers used for quantitative PCR[a]

| Species | Gene | | Primer (5′ to 3′) |
|---------|------|---------|-------------------|
| Cat | TNF-$\alpha$ | Forward | CACATGGCCTGCAACTAATCA |
| | | Reverse | CAGCTTCGGGGTTTGCTAC |
| | IL-6 | Forward | GACTCCAGCCATGACCTTCC |
| | | Reverse | GGGTAGGGAAAGCAGTAGCC |
| | IL-10 | Forward | AAACAGCACGTGAACTCCCT |
| | | Reverse | AGAAATCGATGACAGGCGCC |
| | IL-1$\beta$ | Forward | AACCAACAAGTGGTGTTCCG |
| | | Reverse | GTAGGGTGGGTTTCCCGTCT |
| | IL-18 | Forward | TGACTGTACAGATAATGCACCCC |
| | | Reverse | GCCAGACCTCTAGTGAGGCTA |
| | NLRP3 | Forward | GAGGAGGAAGAGGAGGAGGAAGTG |
| | | Reverse | AAGGCTAACAGTGAGATGGCAGTTC |
| | Caspase-1 | Forward | TCAGGAGGAGGGCTGGTCTA |
| | | Reverse | TGTTTCACCACCTCGTATCCC |
| | JAK2 | Forward | CGAGACCCGACACAGTTTGAAGAG |
| | | Reverse | CACATCTCCACACTGCCAAAATTGC |
| | STAT3 | Forward | GAGAAGGACATCAGCGGCAAGAC |
| | | Reverse | GAGATAAACCAGCGGAGACACGAG |
| | $\beta$-Actin | Forward | CCTGGCACCTAGCACAATGA |
| | | Reverse | CCTGCTTGCTGATCCACATC |

[a]IL, interleukin; TNF-$\alpha$, tumor necrosis factor-$\alpha$.

electrophoresis. *Pfu* high fidelity DNA polymerase was used for PCR amplification, and the number of amplification cycles was strictly controlled. The amplified products were purified and recovered by magnetic beads. The recovered effects were amplified by PCR and quantified by fluorescent light. The fluorescent reagent was Quant-iT PicoGreen dsDNA assay kit, and MiSeq was used to perform high-throughput sequencing. Analysis was performed using QIIME2 software. The $\alpha$-diversity in each sample was assessed based on the distribution of the amplicon sequence variants (ASV)/OTU in the different models. The Kyoto Encyclopedia of Genes and Genomes (KEGG) pathway enrichment analysis was conducted using tools available in the KEGG database. The OTUs reached a 97% nucleotide similarity level.

**Quantitative reverse transcription PCR.** Total RNA was extracted from colon tissue and reversing transcription were performed. The cDNA was synthesized using standard procedures. The primers used to amplify related genes are designed by the software Primer 5.0. The mRNA expression of colonic TNF-$\alpha$, IL-6, IL-10, IL-1$\beta$, IL-18, NLRP3, Caspase-1, JAK2, STAT3, MLC, MLCK, and calmodulin (CaM) was quantified by applying Light Cycler technology and SYBR green PCR core reagent kits. The relative changes in the gene expression were analyzed according to the $2^{-\Delta\Delta CT}$ method. Primers used for quantitative PCR (qPCR) in Table 1.

**ELISA.** Cat colon tissues were homogenized in ice-cold phosphate-buffered saline (PBS) and centrifuged at $13,000 \times g$ at 4°C for 20 min. The supernatant was collected to analyze the concentrations of secretory immunoglobulin. IL-6, IL-1$\beta$, and TNF-$\alpha$ in the serum were measured with the corresponding enzyme-linked immunosorbent assay (ELISA) kits (Nanjing Jiancheng Bio, Inc., China) and measured using a microplate reader (Bio-Rad, Hercules, CA, USA) at 450 nm. All steps were completed according to the manufacturer's instructions.

**Cytosolic Ca²⁺ measurements.** Tissue homogenization was done as described above (53). Calcium ions in the sample were combined with Methyl Thyme Aroma Blue (MTA) in an alkaline solution to form a blue complex. We measured the calcium content of the samples according to the calcium standard curve. The expression of $Ca^{2+}$ in the colon tissues was measured with the corresponding kits (Nanjing Jiancheng Bio, Inc., China, C004-2-1). All steps were completed according to the manufacturer's instructions.

**Western blotting.** The colon tissue (0.1 g) was ground with liquid nitrogen for Western blot analysis. The protein samples were subjected to SDS-polyacrylamide gel electrophoresis with a concentration of 12%,10%, and 8% separation gels. Then, the proteins were transferred to a polyvinylidene fluoride membrane at 250 mA in a Tris-glycine buffer. Subsequently, the membranes were blocked with 5% fat-dry milk at 37°C for 2 h in the shaker, incubated overnight with diluted primary antibodies against the cat, including NLRP3, ASC, Caspase-1, JAK2, STAT3, smooth muscle contraction-associated speck-like protein containing MLC, p-MLC, CaM, and MLCK at 4°C. The membrane was incubated with secondary antibodies for 2 h (horseradish peroxidase-conjugated goat antirabbit IgG, 1:1,000, CST, 7074). The protein bands were detected and quantified using gel imaging system software (TransGen Biotech Co., Beijing, China). $\beta$-Actin and glyceraldehyde-3-phosphate dehydrogenase (GAPDH) were used as internal references. Antibodies information for Western Blot in Table 2.

**Statistical analysis.** Statistical analyses were performed using GraphPad Prism 8.0.1 software (New York, NY, USA). The data are expressed as means $\pm$ standard error of the mean (SEM). One-way analysis of variance (ANOVA) followed by Tukey *post hoc* analysis was applied when evaluating differences between three groups. $P$ values $< 0.05$ were considered statistically significant.

**TABLE 2** Antibodies required for Western blot[a]

| Name | Catalog no. | Company | Dilution times |
|---|---|---|---|
| $\beta$-Actin | AC026 | ABclonal Technology | 1:1,000 |
| GAPDH | D16H11 | Cell Signaling Technology | 1:1,000 |
| NLRP3 | 13158 | Cell Signaling Technology | 1:1,000 |
| ASC | 67824 | Cell Signaling Technology | 1:1,000 |
| Caspase-1 | 24232 | Cell Signaling Technology | 1:1,000 |
| JAK2 | 3230 | Cell Signaling Technology | 1:1,000 |
| STAT3 | 9139 | Cell Signaling Technology | 1:1,000 |
| CaM | 60335to-Ig | Proteintech | 1:500 |
| Rock | 21850-1-AP | Proteintech | 1:1,000 |
| RhoA | 10749-1-AP | Proteintech | 1:1,000 |
| MLCK | 21173-1-AP | Proteintech | 1:500 |
| MLC | 10906-1-AP | Proteintech | 1:500 |
| p-MLC | AF8618 | Affinity | 1:500 |

[a]CaM, calmodulin; GAPDH, glyceraldehyde-3-phosphate dehydrogenase; MLC, myosin light chain; p-MLC, MLC phosphorylation; MLCK, Myosin light chain kinase.

**Data availability.** The data sets supporting the conclusions of this article are available in the NCBI SRA (https://www.ncbi.nlm.nih.gov/sra) under BioProject accession number PRJNA873140.

## SUPPLEMENTAL MATERIAL

Supplemental material is available online only.
**SUPPLEMENTAL FILE 1**, PDF file, 0.2 MB.

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
