## [Reviewer comments · Microbiology Spectrum]

Microbiology Spectrum

Berberine depresses inflammation and adjusts smooth muscle to ameliorate ulcerative colitis of cats by regulating gut microbiota

xuejiao gao, MengYao Guo, xueying li, shuang xu, yanhe zhang, and kan li

Corresponding Author(s): Xue-jiao Gao and Mengyao Guo, Northeast Agricultural University

Review Timeline:

Submission Date:	August 27, 2022
Editorial Decision:	September 10, 2022
Revision Received:	September 20, 2022
Accepted:	September 27, 2022

Editor: Yunhe Fu

Reviewer(s): Disclosure of reviewer identity is with reference to reviewer comments included in decision letter(s). The following individuals involved in review of your submission have agreed to reveal their identity: Jianzhu Liu (Reviewer #2)

Transaction Report:

DOI: <https://doi.org/10.1128/spectrum.03207-22>

September 10, 2022

Prof. xuejiao gao
Northeast Agricultural University
Tiedong Street
Harbin
China

Re: Spectrum03207-22 (Berberine depresses inflammation and adjusts smooth muscle to ameliorate ulcerative colitis of cats by regulating gut microbiota)

Dear Prof. xuejiao gao:

Link Not Available

Sincerely,

Yunhe Fu

Journals Department
Reviewer comments:

Reviewer #1 (Comments for the Author):

This study discussed the changes of intestinal bacteria when colitis occurred. The research in this paper shows that harmful bacteria in the intestinal tract of colitis caused by sodium dextran sulfate increase and beneficial bacteria decrease. Adding berberine can reverse this situation. It is a worthy research direction to study intestinal diseases by detecting the changes of bacterial flora. At the same time, the cat was used as the model animal, and the experimental process was reasonable and novel. And the regulation of inflammatory response by berberine through JAK2 / STAT3 signaling pathway has also been verified, which will contribute to the research of colitis in the future. I suggested to accept for publishing.

1. Line 103 and Line 111, symbols exist in error.

2. Line 106 and Line 109, "DSS-induced" please spell correctly.
3. Line 268, "The fecal samples were collected between groups DSS+BBR and DSS", 16S rRNA gene sequencing study and Bioinformatics Analysis, how many groups of data? Two groups or three groups?
4. Please add the control group information in Fig 2 of figure legends
5. Please specify the linkages between reduced inflammation and smooth muscle in the colon.

Reviewer #2 (Comments for the Author):

UC disease research is very common in human and animals, but there are few studies on cats. This study is of great significance to the development of cat research. As cats live together with humans, some diseases have good comparative medical significance. This is a new direction worth studying. In this paper, high-throughput sequencing was used to determine the bacterial changes in the intestinal tract of colitis. Berberine was found that could alleviate the intestinal flora disorder with dextran sodium sulfate-induced. The present study showed a new direction for the treatment of colitis. It was further verified that berberine could alleviate the inflammatory reaction and regulate the contraction of smooth muscle in intestinal. This paper was interesting. It has a certain guiding significance for the research of non-antibiotic therapy for the treatment of enteritis by relying on the regulation of microbiota. I recommend that this article be accepted for publication.

1. Please add the references in Line 62.
2. Line 67, "BBR had a significant effect on ulcerative colitis bioavailability", What is the bioavailability level?
3. Line 170, "signalling pathway", please spell correctly.
4. How about the successful rate of model construction? Is 18-day sufficient to construct the model?
5. There are still some shortcomings in English writing. It is suggested to modify and perfect it

Staff Comments:

Preparing Revision Guidelines

Please return the manuscript within 60 days; if you cannot complete the modification within this time period, please contact me. If you do not wish to modify the manuscript and prefer to submit it to another journal, please notify me of your decision immediately so that the manuscript may be formally withdrawn from consideration by Microbiology Spectrum.

This study discussed the changes of intestinal bacteria when colitis occurred. The research in this paper shows that harmful bacteria in the intestinal tract of colitis caused by sodium dextran sulfate increase and beneficial bacteria decrease. Adding berberine can reverse this situation. It is a worthy research direction to study intestinal diseases by detecting the changes of bacterial flora. At the same time, the cat was used as the model animal, and the experimental process was reasonable and novel. And the regulation of inflammatory response by berberine through JAK2 / STAT3 signaling pathway has also been verified, which will contribute to the research of colitis in the future. I suggested to accept for publishing.

1. Line 103 and Line 111, symbols exist in error.
2. Line 106 and Line 109, “DSS-induced” please spell correctly.
3. Line 268, “The fecal samples were collected between groups DSS+BBR and DSS”, 16S rRNA gene sequencing study and Bioinformatics Analysis, how many groups of data? Two groups or three groups?
4. Please add the control group information in Fig 2 of figure legends
5. Please specify the linkages between reduced inflammation and smooth muscle in the colon.

Response to Reviewer #1

1. Line 103 and Line 111, the incorrect symbol has been modified.
2. Line 106 and Line 109, "DSS-induced" has been modified to "DSS-induced".
3. Line 268, "The fecal samples were collected between groups DSS+BBR and DSS" has been modified to "The fecal samples were collected between three groups". There are three groups in 16S rRNA gene sequencing study and Bioinformatics Analysis.
4. The control group information has added in Fig 2 of figure legends.
5. According to the contents of the intestine and the external environment, intestinal smooth muscle moves promote the digestion and absorption of food, and helps the excretion of garbage and toxins in the intestine through muscle peristalsis. Colitis causes damage to the gut and changes in the contraction of smooth muscle. The inflammation in the colon is reduced, and the contraction of smooth muscle is also restored.

Response to Reviewer #2

1. The reference has added in Line 62.
2. Line 67, "BBR had a significant effect on ulcerative colitis bioavailability", this refer to the bioavailability of **nutrients** in the intestine that is affected by ulcerative colitis. The bioavailability level indicated as the rate and degree at which a drug is absorbed into the circulation of the body, it describes the percentage of the dose of an oral drug that is absorbed by the gastrointestinal tract and reaches the systemic circulation through the liver.
3. Line 170, "signalling pathway" has been modified to " signaling pathway".
4. The successful rate of model construction is relatively high, there were no dead animals during the experiment. 18-day is sufficient to construct the model, the modeling results are remarkable.
5. English writing has been appropriately rewritten to the best of my ability and underlining the changes in Marked Up Manuscript

September 27, 2022

Prof. xuejiao gao
Northeast Agricultural University
Tiedong Street
Harbin
China

Re: Spectrum03207-22R1 (Berberine depresses inflammation and adjusts smooth muscle to ameliorate ulcerative colitis of cats by regulating gut microbiota)

Dear Prof. xuejiao gao:

Your manuscript has been accepted, and I am forwarding it to the ASM Journals Department for publication. You will be notified when your proofs are ready to be viewed.

Sincerely,

Yunhe Fu
Editor, Microbiology Spectrum

The author has made the revisions as suggested.